# A new dataset for French and multilingual keyphrase generation

**Frédéric Piedboeuf**
RALI, Diro
Université de Montréal
`frederic.piedboeuf@umontreal.ca`

**Philippe Langlais**
RALI, Diro
Université de Montréal
`felipe@iro.umontreal.ca`

## Abstract

Keyphrases are key components in efficiently dealing with the ever-increasing amount of information present on the internet. While there are many recent papers on English keyphrase generation, keyphrase generation for other languages remains vastly understudied, mostly due to the absence of datasets. To address this, we present a novel dataset called Papyrus, composed of 16427 pairs of abstracts and keyphrases. We release four versions of this dataset, corresponding to different subtasks. Papyrus-e considers only English keyphrases, Papyrus-f considers French keyphrases, Papyrus-m considers keyphrase generation in any language (mostly French and English), and Papyrus-a considers keyphrase generation in several languages. We train a state-of-the-art model on all four tasks and show that they lead to better results for non-English languages, with an average improvement of 14.2% on keyphrase extraction and 2.0% on generation. We also show an improvement of 0.4% on extraction and 0.7% on generation over English state-of-the-art results by concatenating Papyrus-e with the Kp20K training set.

## 1 Introduction

As of 2017, approximately 2.5 million new papers are published each year in the scientific field alone[1]. With this ever-increasing flow of information, technologies that attempt to condense information, such as keyphrase generation [33], are becoming increasingly important. Keyphrases are defined as single or multi-word lexical units that summarize documents [15], and they can be used for indexing documents [31], document clustering [20], summarization [40], or even opinion mining [5].

Due to the fast evolution of generative models, keyphrase generation (KPG) has gained attention in recent years. However, most of the models that are available today are trained exclusively in English, despite the fact that English represents only about 25% of internet use[2]. This paper attacks this problem by collecting a multilingual dataset for KPG, composed of mostly French and English documents but occasionally of documents in other languages. To do so, we look at Papyrus[3], the institutional repository of Université de Montréal which hosts theses as well as other types of documents. We collect all documents from Papyrus and separate them into pairs of input sentences (abstracts) and keyphrases.

We split the collected data into four KPG tasks; English, French, as well as two multilingual versions: KPG in a `one2many` way (one abstract in one language generating many keyphrases

---

[1] http://blog.cdnsciencepub.com/21st-century-science-overload/
[2] https://www.statista.com/statistics/262946/share-of-the-most-common-languages-on-the-internet/
[3] https://papyrus.bib.umontreal.ca/xmlui/

36th Conference on Neural Information Processing Systems (NeurIPS 2022) Track on Datasets and Benchmarks.

in that same language), as well as KPG in a `many2many` way (multiple abstracts of the same document generating several keyphrases in various languages). The difference between the two multilingual tasks is illustrated in Table 1. To the best of our knowledge, only English KPG has been presented in the literature as a task so far. We run generative baselines on each task based on the BART architecture [27], which achieved state-of-the-art results in English KPG [9], and we highlight several limitations that these systems have. We also run a version where we concatenate our English version of Papyrus with the main KPG English dataset, Kp20K, and show that it helps outperform state-of-the-art results on English KPG.

The paper is organized as follows. In Section 2, we present the related work that has been done in the field. In Section 3, we present the methodology for collecting the datasets, as well as some analysis of their contents. Sections 4 and 5 demonstrate the efficacy of the models trained on our new datasets and discuss the results on the different tasks. Then in Section 6, we discuss broader implications of our work, and Section 7 concludes the paper.

## 2 Related Work

This work bridges two branches of automatic indexation: KPG in English, which has received a lot of attention in recent years, and keyphrase extraction for other languages. The difference between extraction and generation lies in the fact that extractive methods can only predict keyphrases present in the input (*present keyphrases)*, while generative methods can also predict keyphrases not explicitly written (*absent keyphrases*).

The recent success of English KPG can be mostly attributed to the paper of Meng et al. [34], which proposes the first large corpus for keyphrase generation, Kp20K. The authors collect 567,830 abstracts/keyphrases pairs of scientific articles in computer science, and show that an encoder-decoder architecture pretrained on Kp20K can generate accurate keyphrases for other, small datasets.

Most KPG papers following Meng et al. [34] use Kp20K as a training corpus. In Yuan et al. [46], the authors show that training the system to generate a sequence of keyphrases (`one2many`) is more efficient than training to generate one keyphrase and using a beam-search for multiple keyphrases (`one2one`), because the network could use the information of preceding generated keyphrases for more diversity. This sparked many other works, including the papers of Meng et al. [32, 33], which show that it is more efficient to train to generate first the present keyphrases (those in the input), followed by the absent ones. Other improvements have also been reported, such as the use of reinforcement learning to encourage the generation of more keyphrases [8], the study of generation from longer documents [1; 29], the generation of sets instead of sequences (`one2set`) [45], the use of summarization techniques [48], or the use of GANs for better generation of absent keyphrases [42; 43].

Recently, Chowdhury et al. [9] reviewed papers and commented on the difficulty of implementing the proposed methods. They show that by simply fine-tuning a pre-trained transformer, BART [27], they could achieve results equating state-of-the-art.

Keyphrase extraction has been explored in the past for several languages, mostly with unsupervised approaches and small datasets. Giarelis et al. [17] recently reviewed and compared several multilingual extractive methods on five languages: French, Spanish, Polish, Portuguese, and English.[4] They show that three techniques perform better: YAKE [7], a statistical method relying on several features such as the position of the term or its normalized frequency, KeyBert [19], an extraction method using BERT [10] to get document representation and calculating similarity with n-gram embeddings, and singleRank [44], a modification of TextRank [35] where the weighting is adapted for keyphrase extraction. These three models, as well as the five non-English datasets, will serve to evaluate the efficacy of our generative approach in Section 4.

Other efforts at keyphrase extraction on languages other than English include DEFT2012 and DEFT2016 challenges, where keyphrases extraction and indexation are tested on French articles from the humanities [38; 2; 14]. From what we can infer, this dataset is no longer maintained, and we weren't successful in obtaining it. Many other techniques have been

---

[4]The number of entries in the datasets used by Giarelis et al. [17] ranges from 50 to 100.

developed for keyphrase extraction in other languages, such as [21] where the authors introduce a TF-IDF method tailored for multilingual texts, the development of statistical methods for language independent keyphrase extraction [39], and the development of KPG systems for specific languages [13; 11].

Finally, the recent paper of [16] introduces two datasets for multilingual KPG, one from the e-commerce domain, composed of entries in Spanish, German, Italian, and French, and one in the academic domain, composed of entries in Korean and Chinese. However, their paper presents significant differences with ours. First, the focus is on a new technique they name *Retriever-Generator Iterative Training*, where the examples are augmented with English keyphrases retrieved from semantically close examples. Second, while they show the efficacy of their technique on their dataset, we focus on showing that having a multilingual corpus to pretrain a KPG system leads to downstream improvements on various test datasets from different languages and domains, which can be more generally useful to practitioners. While we wanted to compare their dataset to ours, we were unfortunately unable to access their datasets at the time of writing.

# 3   Data collection

We collect our multilingual corpus from Papyrus, which is a repository of documents from the Université de Montréal. Papyrus hosts various types of documents, mostly theses but also institutional reports or other documents written by faculty members. While official policy is that theses should be written in French, most have abstracts in several languages, mainly French and English. For the remaining of this paper, we'll denote *document* an entry as it appears in Papyrus, with all its abstracts and keyphrases.

The documents can be accessed by their index, so we could easily scrape the 26508 pages that were available at the date of collection.[5] Of these, 657 corresponded to error pages, and 9602 to documents with no abstracts or keyphrases, leaving us with a total of 16249 documents composed of one or more abstracts as well as multiple keyphrases. The next step is then to conduct language identification on both the abstracts and keyphrases.

We use `langdetect`[6] on the abstracts, and a simple heuristic on the keyphrases to find their languages, among all the languages identified from the abstracts of that document[7]: for present keyphrases, we assign them to all the languages of the abstracts that contains them. For absent keyphrases, we use `fastText` [23; 24] and assign them to the most likely language. If `fastText` fails to identify the language, that is, if the top-15 languages from `fastText` are not concordant with the languages of the abstracts, such as for the keyphrase "間 ma", "1000", or "Leptoquark", we assign it to all possible languages for that entry. This corresponds to 119 keyphrases out of 208730. This approach is not perfect, as some keyphrases are sometimes erroneously tagged, and so as a sanity check we inspected 100 randomly selected examples from the training set and checked whether the abstracts and keyphrases are labelled with the correct language. The sanity check was done by the first author who speaks French, English, Spanish, and Dutch. We measure an accuracy of labelling of 100% for the abstracts, and 98.9% for the keyphrases. The selected examples and results of the evaluation are available on github.[8]

Finally, we split the entries into four different KPG datasets/tasks and drop examples that are present more than once:

- **Papyrus-f**: From the French abstracts, generate French keyphrases,
- **Papyrus-e**: From the English abstracts, generate English keyphrases,
- **Papyrus-m**: From one abstract in any language, generate keyphrases in that same language (one language to one language),

---

[5] The latest version of the dataset was collected on April 7, 2022.

[6] `https://pypi.org/project/langdetect/`

[7] During the language identification step, we also filter out two types of keyphrases that are standard classification corresponding to the UMI and JEL standard. We filter them out because they are not universally used and correspond to the task of indexation rather than KPG.

[8] All our code for preprocessing and running experiments is available at `https://github.com/smolPixel/French-keyphrase-generation`. The data is distributed with the authorization from Université de Montréal.

- **Papyrus-a**: From the multiple abstracts of a document, generate keyphrases in the same languages as the abstracts (many to many languages).

We show a toy example in Table 1. For Papyrus-a, we simply concatenate the abstracts together in the order they were present in Papyrus, and we do the same for keyphrases.

| Task | Abstract | Keyphrases |
|---|---|---|
| Papyrus-f | Ce document parle de l'extraction et de la génération de mots-clés francophone et multilingue. | Extraction de mots-clés, génération de mots-clés, tâche francophone, tâche multilingue |
| Papyrus-e | This document is about keyphrase generation and extraction, for French and multilingual corpora. | keyphrase generation, keyphrases extraction, Multilingual keyphrases, document indexing. |
| Papyrus-m (entry 1) | Ce document parle de l'extraction et de la génération de mots-clés francophone et multilingue. | Extraction de mots-clés, génération de mots-clés, tâche francophone, tâche multilingue |
| Papyrus-m (entry 2) | This document is about keyphrase generation and extraction, for French and multilingual corpora. | keyphrase generation, keyphrases extraction, Multilingual keyphrases, document indexing. |
| Papyrus-a | Ce document parle de l'extraction et de la génération de mots-clés francophone et multilingue. This document is about keyphrase generation and extraction, for French and multilingual corpora. | Extraction de mots-clés, génération de mots-clés, tâche francophone, tâche multilingue, keyphrase generation, keyphrases extraction, Multilingual keyphrases, document indexing. |

Table 1: A toy example of a bilingual document and how it is processed for the various tasks. The present keyphrases are underlined. For Papyrus-m, each language is separated and passed as input separately, while in Papyrus-a the input text is a concatenation of all of the abstracts of that document.

We separate the documents with a 70/10/20 ratio train, dev, and test, prior to separating the dataset into the four tasks, so that the same documents find themselves in the same split for each task. We report in Table 2 the number of examples in each split and for each task. If we look at the number of languages in each document, we find 1761 unilingual documents, 9391 bilingual ones, 134 with three languages, three with four, and one with six languages.

If we look instead at the languages present in the dataset, we find 15289 English abstracts, 14826 in French, 172 Spanish, 29 German, 20 Italian, 17 Portuguese, 7 Arabic, 5 Tagalog, 3 Catalan/Greek, 2 Turkish/Russian, and 1 in Polish/Farsi/Indonesian/Lingala/Swedish/Finnish/Romanian/Korean. There are also two abstracts written in a language that `langdetect` could not identify. One is, as far as we can judge, Inuktitut. The other was a mistaken entry where, in addition to the French and English abstracts, the term "2002-10" had been marked as an abstract.[9]

Table 3 presents some statistics about the abstracts and keyphrases in our datasets as well as for Kp20K. The abstracts in Papyrus are longer than those in Kp20K, with more keyphrases, and a larger proportion of present keyphrases.

| Dataset | #Train | #Dev | #Test |
|---|---|---|---|
| Papyrus-f | 10299 | 1488 | 2981 |
| Papyrus-e | 10508 | 1539 | 3046 |
| Papyrus-m | 20963 | 3040 | 6061 |
| Papyrus-a | 11290 | 1638 | 3261 |

Table 2: Number of examples in each split and for each dataset.

---

[9]To make sure that no other artifacts were introduced, we sorted the entries by length and manually checked them. The next shortest entry after "2002-10" is "Éditorial en réponse à un article d'Andrée Quiviger.", which is a correct abstract for the document.

| Dataset | len.abstract | #kp | len.kp | % present kp | % br.present kp |
|---|---|---|---|---|---|
| Kp20K | 148 | 5.3 | 2.1 | 50.9 | 60.8 |
| Papyrus-f | 323 | 7.0 | 1.9 | 65.0 | 72.2 |
| Papyrus-e | 290 | 7.4 | 1.7 | 60.8 | 67.6 |
| Papyrus-m | 307 | 7.2 | 1.8 | 62.8 | 69.8 |
| Papyrus-a | 573 | 13.4 | 1.9 | 62.8 | 70.4 |

Table 3: Statistics of various tasks. Values represent the average over all examples of the training sets. *br.present kp* indicates "broken" present keyphrases, that is, keyphrases for which each word are individually present in the abstract.

## 4 Baselines

Our work allows to train large abstractive models for French and Multilingual KPG. Until now, the only option was to use unsupervised, language independent, extractive approaches. In this section, we first train four models, for the four tasks, and run them on their own test sets. Then, we compare their performances on various KPG tasks in several languages and show that using pretraining is more efficient than unsupervised approaches.

We follow the footsteps of Chowdhury et al. [9] in opting for a simple approach that is efficient and reproducible. We extend their proposed method, which uses BART for KPG, to a multilingual setting. Similarly to them, we apply no special preprocessing to the sentences, and do not order keyphrases in any specific way. Because BART is a model trained on English sentences, it would not work optimally on Papyrus-f, Papyrus-m, and Papyrus-a. As such, we use BARThez [12] (a French version of BART) for Papyrus-f, and the multilingual BART model mBART-large-50 [28] for Papyrus-a and Papyrus-m. We also rerun and report performances on Kp20K for comparison. It is to note that for mBART-large-50, the language has to be specified when tokenizing. For languages that are not available with mBART-large-50, or for the multilingual inputs of Papyrus-a, we default to English. For the rest of this paper we denote the BART systems fine-tuned on Papyrus-f, Papyrus-e, Papyrus-m, Papyrus-a, and Kp20K, as respectively Bart-f, Bart-e, Bart-m, Bart-a, and Bart-k. Finally, we include in the Appendix results on T5 [41], which has shown good results in tasks like summarization [18]. However, we found it underperformed in the KPG task when compared to BART.

We use a learning rate of 5e-5 with the AdamW optimizer, a beam search of 10 for generation, and a maximum input length of 1024. The only notable difference with Chowdhury et al. [9] is that we emulate a batch size of 128 by using a small batch size and gradient accumulation so that it fits on our GPU, while they report directly using a batch size of 128. We also use 16-bits precision, and train for 10 epochs for all Papyrus datasets, which seems to give optimal results, and 3 for Kp20K, which is the value used in Chowdhury et al. [9]. All experiments are run on a computer with the following configuration - *Processor*: AMD Ryzen 7 5800X 8-Core Processor @ 3.8GHz *RAM*: 126G *GPU*: GeForce RTX 3090, 24 G.[10] We run all experiments three times and report the average and standard deviations.

It is to note that in the case of Papyrus-m, we have a skewed distribution, with most entries being in French or English, and only a minority being in other languages. Because it is so extreme, we decided not to make use of it while training nor change the evaluation protocol. We discuss in further details the impact of the skewed distribution in Section 5.

We use the evaluation protocol from [46; 34; 32], which calculates recall, precision, and F1, for all keyphrases present in the abstract, as well as for the absent ones. Typical evaluation in KPG is done at thresholds where we take the $X$ best keyphrases generated, denoted $M@X$, where $M$ is the metric of interest. We report Precision, Recall, and F1@5 and @10 for present keyphrases, and @10 for absent keyphrases, similarly to Chowdhury et al. [9], but remove the stemming process in the evaluation to accommodate for non-English languages.

Table 4 shows the results for present and absent keyphrases on all five datasets for the system trained on that dataset. We see that the results are comparable to state-of-the-art ones, at least on Kp20K where Chowdhury et al. [9] reported 33.5 F@5/31.1 F@10 for present

---

[10] Fine-tuning time (without testing) is 1h for Bart-f, 2h for Bart-e, 8h for Bart-m, 5h for Bart-a, and 15h for Bart-k.

| | Present | | | | | | Absent | | |
|---|---|---|---|---|---|---|---|---|---|
| | **P@5** | **R@5** | **F@5** | **P@10** | **R@10** | **F@10** | **P@10** | **R@10** | **F@10** |
| Kp20K | 29.5 | 37.1 | 30.4 | 29.5 | 37.2 | 30.4 | 8.8 | 5.0 | 5.8 |
| Papyrus-f | 31.5 | 24.8 | 25.6 | 31.4 | 25.0 | 25.7 | 4.8 | 2.8 | 3.1 |
| Papyrus-e | 35.5 | 37.4 | 34.1 | 34.6 | 39.8 | 34.6 | 7.7 | 4.3 | 4.9 |
| Papyrus-m | 33.4 | 31.5 | 30.3 | 33.1 | 32.8 | 30.6 | 5.7 | 3.4 | 3.8 |
| Papyrus-a | 30.6 | 17.5 | 20.1 | 30.1 | 20.6 | 21.9 | 6.0 | 3.0 | 3.5 |

Table 4: Evaluation of the generative models (respectively Bart-k, Bart-f, Bart-e, Bart-m, and Bart-a) on their own test sets. In listed order, the standard deviations for each model are in the range [0.4, 0.8], [0.2, 0.9], [0.1, 0.9], [0.7, 4.0], [1.4, 4.9].

| Test set \ Model | **SR** | **YAKE** | **KB** | **Bart-f** | **Bart-e** | **Bart-m** | **Bart-a** | **Bart-k** |
|---|---|---|---|---|---|---|---|---|
| Papyrus-f | 6.6/0 | 8.1/0 | 2.4/0 | 25.7/2.8 | 24.8/0.7 | **30.4/3.6** | 24.0/2.8 | 18.2/0.3 |
| Papyrus-e | 8.6/0 | 13.2/0 | 2.2/0 | 19.2/0.2 | **34.6/4.3** | 30.9/3.2 | 23.4/2.2 | 31.2/2.0 |
| Papyrus-m | 8.2/0 | 11.8/0 | 2.5/0 | 22.3/1.5 | 29.6/2.5 | **30.6/3.4** | 23.7/2.5 | 24.7/0.4 |
| Papyrus-a | 1.4/0 | 9.7/0 | 1.9/0 | 17.2/1.3 | 22.0/1.9 | **22.6**/2.5 | 21.9/**3.0** | 17.6/1.0 |
| Wikinews (fr) | 14.2/0 | 23.9/0 | 6.0/0 | 23.1/0.3 | **27.4**/0.4 | 26.4/**0.5** | 18.1/**0.5** | 22.2/0.4 |
| cacic57 (es) | 0.3/0 | 11.8/0 | 5.2/0 | 29.1/0 | 47.5/0 | **48.8**/0 | 35.2/0 | 45.6/0 |
| wicc 78 (es) | 0.4/0 | 11.5/0 | 2.7/0 | 14.6/0 | 26.4/0 | **30.1**/0 | 23.6/0 | 29.9/0 |
| 110ptbnkp (pt) | **21.4**/0 | 19.3/0 | 4.9/0 | 5.5/0 | 18.1/0 | 10.3/0 | 8.6/0 | 15.0/0 |
| pak2018 (pl) | 0.7/0 | 0.7/0 | 0.8/0 | 2.6/0 | 6.0/0.2 | **7.4/4.6** | 6.2/1.9 | 5.1/0 |
| Average | 6.8/0 | 12.2/0 | 2.1/0 | 17.7/0.7 | 26.3/1.1 | **26.4/2.0** | 20.5/1.4 | 23.3/0.5 |

Table 5: Evaluation of the systems on various test sets. We report F1@10/R@10 for present and absent keyphrases respectively. SR stands for SingleRank, and KB for KeyBERT. Bart-a, Bart-f, Bart-e, Bart-m, and Bart-k correspond to our Bart models trained with respectively Papyrus-a, Papyrus-f, Papyrus-e, Papyrus-m, and Kp20K. Best scores for each test set are in bold.

keyphrases and 6.1 R@10 for absent ones.[11] We observe that the performance on Papyrus-f is much lower than for Papyrus-e and -m. As we discussed in Section 5, this is because BARThez underperforms globally and not because the task is more difficult, as we find that training mBART-large-50 on Papyrus-f yields better results. We also note that while Papyrus-m and Papyrus-a correspond to two versions of the same task, the performance on Papyrus-a is lower. This suggests that the `many2many` setting is more difficult than the `one2many` one, and that further research on the subject is necessary. Finally, we observe that the standard deviation is higher for Bart-m and Bart-a, the two multilingual systems using mBART-large-50. While fine-tuning, we noticed that mBART-large-50 was more difficult to correctly fine-tune, sometimes randomly collapsing and getting 0 on all metrics.

We now check if our models help in downstream KPG. We take as our starting point the paper of Giarelis et al. [17] which compares multiple keyphrase extraction methods on datasets of various languages. We report in Table 5 the results for all five BARTs trained on the different tasks as well as the top three systems from Giarelis et al. [17], that is SingleRank, YAKE, and KeyBERT, on the same five non-English datasets they use. Standard deviations follow the tendencies expressed in Table 4.

The French dataset is WikiNews [6], composed of 100 news articles annotated by students. For Spanish, they use subsets of Cacic and Wicc, two datasets composed of scientific articles as well as their keyphrases [3]. For Polish, they use pak2018, which is also composed of abstracts of scientific articles [7], and the Portuguese one is 110ptbnkp, composed of transcribed texts from broadcast news programs [30]. Despite some of these tasks having a low number of present keyphrases, they were all previously considered only for extractive techniques.[12]

---

[11] The lower performance is in part due to the removal of the stemming step in the evaluation protocol which will not accept variations of the same word as one, and in part due to the difference due to randomness of the training process. As far as we could gather, Chowdhury et al. [9] runs only one experiment, while we average over three. When we evaluate with the stemming step, we obtain 33.2 F1@5 and 33.3 F1@10 present, and 3.4 R@10 absent.

[12] They have respectively 94.5%, 70.8%, 62.6%, 15.4%, and 97.8% of present keyphrases.

We note that except for 110ptbnkp, using a generative model allows for a better performance for both present and absent keyphrases. The performance on the absent keyphrases is still low, but better than the extractive systems that cannot generate new keyphrases. It is both surprising and encouraging however to notice that the generative systems outperform generally the extractive systems even on present keyphrases. We also see that Bart-m is the best system overall, surpassing the other systems on 10 out of the 18 results reported. This is not surprising, given the multilingual aspect of the task, and demonstrates the advantage of having a multilingual corpus to train on. Finally, while Papyrus-f and Papyrus-a are useful for objective comparison, they are of limited use for pretraining, Bart-f and Bart-a being vastly outperformed by the other models. For Papyrus-f, it seems to be due to the weak general performance of BARThez, which is also supported by the fact that Bart-e outperforms Bart-f on Wikinews. For Papyrus-a, it calls for further research on the `many2many` setting.

## 5 Discussion

In this section, we analyze the results from three points of view. We first take a look at French KPG, then multilingual KPG, and finally we look at the difference between our dataset and Kp20K. We conclude by showing an example of an input and the keyphrases generated by each of the generative models.

### 5.1 French keyphrase generation

As mentioned, we believe that the poor performance of Bart-f when compared to Bart-e or Bart-m is due to the poor performance of Barthez rather then a more difficult task. To see if we could get a better system for French KPG, we train two other systems on papyrus-f: mBARThez, which consists in the mBART system that was further trained on a French corpus, as well as mBART. We report results on our two French corpus in Table 6. We can see that with these systems, performance is equivalent and even surpassing Bart-m, confirming that the problem was that BARThez underperforms. The good performance on Wikinews also shows that our systems can be used for downstream KPG in French.

|           | BARThez  | mBARThez  | mBART    | Bart-m   |
|-----------|----------|-----------|----------|----------|
| Papyrus-f | 25.7/2.8 | **32.4/4.0** | 31.6/3.8 | 30.4/3.6 |
| Wikinews  | 23.1/0.3 | **28.6**/0.2 | 28.5/**1.2** | 26.4/0.5 |

Table 6: Results for three systems trained on Papyrus-f on the two French corpus, compared to Bart-m. We can see that when mBARThez and mBART are used, systems trained on Papyrus-f outperform Bart-m.

### 5.2 Multilingual keyphrase generation

As mentioned, the best overall system for multilingual KPG is Bart-m. Some surprising results are that Bart-e (and to some extent, Bart-f) can successfully extract keyphrases from corpora in other languages. By looking at 100 keyphrases correctly predicted by Bart-e on Papyrus-f, we found 39 which could be considered anglophones (*gpr55*, *queer*, etc.), and the rest were clearly francophone (*imaginaire social*, *coopération réglementaire*, etc.), supporting the hypothesis that the models use spurious cues for KPG.

|        | fr   | en   | es   | pl  | it   | de   | pt   | tl   |
|--------|------|------|------|-----|------|------|------|------|
| Bart-a | 9.2  | 10.2 | 12.7 | 8.3 | 13.7 | 15.1 | 15.0 | 0.0  |
| Bart-m | 12.0 | 15.3 | 20.7 | 2.8 | 23.5 | 29.0 | 45.0 | 11.1 |

Table 7: R@10 for both present and absent keyphrases and for each language present in respectively the test sets of Papyrus-a and Papyrus-m. Language codes follow the ISO 639-1 format, with fr being French, en-English, es-Spanish, pl-Polish, it-Italien, de-German, pt-Portuguese, and tl-Tagalog.

To better appreciate the performances of our multilingual systems, we report in Table 7 the R@10 by languages for Bart-a and Bart-m, which indicates how many of the keyphrases of

|  | Bart-f | Bart-e | Bart-m | Bart-a | Bart-k | Bart-(k+e) |
|---|---|---|---|---|---|---|
| Inspec | 21.1/0 | 27.8/1.1 | 20.4/0.8 | 18.9/0.5 | **28.5**/1.3 | 27.8/**1.4** |
| NUS | 19.2/0 | 31.3/2.4 | 26.4/0.8 | 22.2/0.5 | **35.7/3.0** | 35.5/2.8 |
| SemEval | 15.3/0.1 | **22.9**/0.9 | 19.4/0.5 | 16.2/0.2 | 20.7/1.2 | 20.8/**1.6** |
| Kp20K | 13.3/0.1 | 23.2/2.1 | 20.0/1.2 | 17.2/1.0 | 30.4/**5.0** | **30.5**/4.9 |
| Krapivin | 11.2/0.1 | 23.8/2.0 | 18.8/1.3 | 15.9/1.1 | 27.4/**5.8** | **28.0**/5.6 |
| Papyrus-e | 19.2/0.2 | **34.6**/4.3 | 30.9/3.2 | 23.4/2.2 | 31.2/2.0 | 33.9/**6.2** |
| Average | 14.9/0.1 | 27.3/2.1 | 22.7/1.3 | 19.0/0.9 | 29.0/3.1 | **29.4/3.8** |

Table 8: F1@10 present/R@10 absent, on all English test sets for all the generative systems.

each language are found by the systems. We can see that Bart-m performs much better on the test set than Bart-a, being able to generate keyphrases for most of the rare languages. In this specific case, we could elect to simply split by languages and generate keyphrases that way, but there are many contexts where two or more languages might be used in a random order in a text (due to multilingual writers using "code-switching" [37]), and the Papyrus-a task reflects that phenomenon.

Something that is noteworthy is that even if the abstractive models perform globally better, they still struggle with their generative capabilities. A huge part of the problem is the evaluation protocol, which at the moment demands exact match for an absent keyphrase to be deemed correct. However, for a same concept there are many valid ways to express it, and authors do not write all the possible keyphrases that could be a valid match. As a sanity check that the poor performance on absent keyphrases is not due to the models collapsing to pure extraction, we measure what percentage of keyphrases are absent keyphrases. Bart-f outputs 24% of absent keyphrases, Bart-e, 19.8%, Bart-m, 21.3%, Bart-a, 27.2%, and finally Bart-k outputs 23.2% of absent keyphrases. This confirms that the systems do learn to generate new keyphrases, but these keyphrases are simply not evaluated by the current evaluation protocol as correct.

Finally, an important factor to consider for Bart-m and Bart-a is the skewed distribution. Having such a distribution is generally problematic, as deep learning models cannot learn correctly from classes for which there are few data points. Standard solutions include data preprocessing such as synthetic data generation or upsampling, or algorithm approaches such as cost-sensitive learning. However, these are often ineffective when dealing with such a skewed distribution [47]. Nevertheless, our results in Table 7 show that Bart-m and Bart-a correctly learn to generate keyphrases independently of the language.

## 5.3   Difference with Kp20K

We compared, up to now, our new datasets to Kp20K in an informal way, using it as a way to validate our results against state-of-the-art English KPG. In this section, we take a closer look at the performance of the models trained on our new datasets, on English KPG. We report in Table 8 the performance of our models on all the test sets used in Chowdhury et al. [9], namely Inspec [22], NUS [36], SemEval [25], Kp20K [34], and Krapivin [26].

It is surprising that Bart-e is not far behind Bart-k, because Kp20K contains 527,091 data points, compared to the 10,602 examples of Papyrus-e.[13] This prompts us to train a version of BART where we extend the Kp20K dataset with Papyrus-e: Bart-(k+e). Bart-(k+e) overall provides a slight improvement in performance in both extractive and abstractive capacities, surpassing Bart-k by an average of 0.4% on extraction and 0.7% on abstraction.

It is also surprising that Bart-f obtains decent scores for present keyphrases, given that all the test sets are in English. This, combined with results in the preceding section, reinforce the idea that BART generally relies on spurious cues for its KPG instead of on a real comprehension of the task.

---

[13]If we train on only 10,602 points of Kp20K, we get present/absent score of 26.3/1.3 for Inspec, 31.4/2.1 for NUS, 21.8/1.5 for SemEval, 25.3/2.8 for Kp20K, 25.7/3.4 for Krapivin, and 28.5/1.9 for Papyrus-e.

Figure 1: Example of a document from the dev set. We only show here the English abstract, but it is available in English, French, and Spanish. We show the French keyphrases in green, the English ones in red, and the Spanish ones in blue.

**Title: Challenges to reintegration : barriers to reentry encountered by ex-convicts in halfway houses**

**Abstract:** Reintegration is a difficult process and individuals who have been affected by the justice system can benefit from interventions that help them to be autonomous, law-abiding, and integrated into their communities. Current research, however, seldom goes beyond identifying the difficulties in this process. Our qualitative study was an in-depth exploration of the challenges encountered by ex-convicts who had been provided with lodging in halfway houses as part of their reentry process. Six categories of challenges were identified following a content analysis of the interviews (n=16) : return to society, financial and occupational situation, social network, personal development, stigmatization, and living in halfway houses. Experiences specific to certain profiles (length of sentence, sexual delinquency) are noted. Results are discussed in relation to desistance theories. Among other things, the study highlights the ways in which legal conditions and the stigmatization of offenders can lead to counter-productive outcomes.

**Keyphrases:** Réinsertion sociale, Interventions, Libération conditionnelle, Entretiens semi-dirigés, Délinquance sexuelle, Reentry, Parole, Intervention, Semi-structured interviews, Sex offenders, Reinserción social, Intervenciones, Liberación condicional, Entrevistas semi-directivas, Delincuencia sexual

## 5.4 Examples of generated keyphrases.

We show in Figure 1 an example of a document from the development dataset. The abstracts are available in English, French, and Spanish, but for reasons of space we only show the English one. In Table 9, we show the generated keyphrases by all five generative models. All of them are quite good, but generate ultimately few keyphrases. Furthermore, we also point out that Bart-a, trained on the Papyrus-a task, did not generate any Spanish keyphrases, which shows again that the Papyrus-a task is a more demanding setting than Papyrus-m and could benefit from more research.

| | |
|---|---|
| Bart-e | Reintegration, Desistance, Criminal reentry, Halfway house, Stigmatization |
| Bart-f | réinsertion sociale, retour à la liberté, situation financière et occupationnelle, réseau social, stigmatisation, hébergement en maison de transition |
| Bart-m (f) | Désistement criminel, Intervention, Réinsertion sociale, Stigmatisation |
| Bart-m (e) | Stigmatization, Return to society, Social network, Personal development, Sexual delinquency, Halfway houses |
| Bart-m (es) | reinserción social, red social, desistimiento delictivo, estigmatización, desafíos contra productivos |
| Bart-a | Désistement criminel, Hébergement en maison de transition, Stigmatisation, Retour à la société, Réinsertion sociale, Criminal desistimiento, Halfway houses, Return to society, Social reintegration, Stigmatization |
| Bart-k | reintegration, justice system, desistance theory, halfway houses |

Table 9: Keyphrases produced by each of the generative models for the example of Figure 1. In this case, all generative systems only returned present keyphrases.

# 6 Implications

We do not see any direct negative implications of the new tasks and datasets, and rather we believe that multilingual datasets are a good thing that allow for a fairer use of artificial intelligence [4]. There is, however, the danger of misinterpreting the results, in part due to the skewed distribution. The classic evaluation metrics do not really reflect the performance on the languages that are rare in the dataset, so researchers should take care to interpret correctly the performance lest it erases the performance on minority languages. We believe that the best way to evaluate the performance on papyrus-m is to test on external test sets on other languages as well as output the performance per language to verify that it performs well on all languages.

# 7 Conclusion

Keyphrase generation is an important topic for machine learning, due to the increasing amount of information that is accessible on the internet. Until now, however, research has mostly been focussed on English KPG, which represents only a subset of the need for KPG.

In this work, we first collect and curate a dataset for multilingual KPG, which we separate into four distinct tasks, among them three that have never been presented before. Papyrus-f and Papyrus-e are unilingual KPG tasks, in French and English respectively. Papyrus-m and Papyrus-a are both multilingual KPG tasks, the former in a one-to-one way and the later in a many-to-many way.

We train a state-of-the-art KPG model based on BART, on all four datasets and test the models on test sets from various languages. We show that while BART provides a solid baseline, there is still a lot of work that remains to be done.

In particular, most universities over the world maintain a repository of theses similar to Papyrus.[14], and our work demonstrates that this is useful data that should be mined for better multilingual KPG. We hope this work will encourage others to do so and will help kickstart multilingual KPG, which is without a doubt a very important field in modern machine learning.

## Acknowledgments

We thank LexRock[AI], MITACS, Université de Montréal, and the Papyrus team for supporting this work, as well as the six reviewers of this paper for their insightful comments.

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
