# OpenReview forum: "A new dataset for multilingual keyphrase generation"
_NeurIPS.cc/2022/Track/Datasets_and_Benchmarks — NeurIPS 2022 Datasets and Benchmarks _

### Official Review · Reviewer_2e1j · 2022-07-26
**Review of "A new dataset for multilingual keyphrase generation"**

**Rating:** 6
**Confidence:** 3

**Strengths:**

- well-written paper
- very good expansion to English only keyword extraction datasets that could pave the way for using other universities' libraries to get even more linguistic diversity
- evaluation conducted on various languages


**Weaknesses:**

- "Multilinguality" is in practice limited to English and French, as the third most common language (Spanish) has only 172 items

- What is the purpose of introducing the long tail of languages, especially when there are only one or two occurrences? It can introduce a bias in evaluations

- It would be nice to see some comparison of other models on Kp20K and other datasets in Table 7, especially if they are available in previous work: https://arxiv.org/abs/2201.05302


**Additional Feedback:**

It would be nice to see the distribution of labels by the field of study to check if some disciplines are not too much over/underrepresented, so the model will not learn the artefacts.


**Clarity:**

The paper is fairly clear and easy to understand.

One suggestion that could make the flow even better would be to consider moving some of the footnotes into the main text.

**Correctness:**

- **L146-148**: "It is to note that for mBART-large-50, the language has to be specified when tokenizing. For languages that are not available with mBART-large-50, or for the multilingual inputs of Papyrus-a, we default to English. "

---> I don't understand this part. Does it mean that for Papyrus-a (EN+FR+others), the only tokenizer used was an english tokenizer? What was the rationale for this choice when half of the dataset contains french keywords and abstracts?



- **L127-128**: "One is, as far as we can judge, Inuktitut. The other was a mistaken entry where, in addition to the French and English abstracts, the term ”2002-10” had been marked as an abstract."

---> Maybe it makes sense to add a threshold on the minimum text size as too short texts can generate noise?


- Having introduced the Papyrus-m version of the dataset, what are the gains of also having Papyrus-a? Wouldn't it only introduce noise and misclassification about their usage?

**Documentation:**

- Preprocessed dataset is available on Github in a tsv format, easy to load and run experiments
- Scripts for preprocessing datasets and training models are available
- It is not authors fault, but I couldn't access the Papyrus repository: https://papyrus.bib.umontreal.ca/xmlui/


**Ethics:**

No ethical concerns

**Relation To Prior Work:**

Related work is clearly discussed.

Just pointing as an additional reference very recent paper proposing a similar dataset for six languages (although I was not able to access the dataset itself, so it can be hard to compare):
Retrieval-Augmented Multilingual Keyphrase Generation with Retriever-Generator Iterative Training
Yifan Gao, Qingyu Yin, Zheng Li, Rui Meng, Tong Zhao, Bing Yin, Irwin King, Michael R. Lyu
https://arxiv.org/abs/2205.10471

**Summary And Contributions:**

This paper introduces a new dataset for multilingual keyphrase generation called Papyrus. The dataset contains 16k pairs of abstracts and keyphrases essentially for English and French with a long tail of additional languages that, in practice, can only be used for evaluation. The authors release four versions of this dataset with different language configurations. Using multilingual Papyrus, they train the BART model, which shows better results for French, Spanish and Polish keyword extraction datasets. Training BART on the Kp20K and English version of the Papyrus dataset exhibits even better performance on a set of benchmarks from https://arxiv.org/abs/2201.05302. The percentage of absent keywords is lower than in Kp20K, but considering the dataset size can be helpful for training models.

---

> ### Author Response · Authors · 2022-08-10
> **Response to "Review of "A new dataset for multilingual keyphrase generation" " (2)**
>
> Thank you for your time and thoughtful comments.
> -The long tail of languages was kept as a design choice because 1- it reflects real world academic data, 2- it is easily removable from our dataset if someone would wish to use the dataset without it, 3- They are useful for evaluating how multilingual KPG does and allow to point out some interesting characteristic of the system 4- Even if we expend the dataset to cover more languages, some are always going to be present in small quantities, such as Inuktitut, and we don't believe it is worth to cut them out as it would simply further the problem of these languages not being represented in datasets.
> -We are also currently working on more models that we hope we will have results for  in the final submission (see official comment on the submission).
> - Yes, for papyrus-a, in case more than one language was used, the English tokenizer was used. If the entry is only in French, then the French tokenizer is used. The rationale was that given a multilingual entry, a choice has to be made of tokenizer, and since English tools are often better than their French counterpart, we picked English. While this is a problem related to the use of bartm, we believe it shows the use of papyrus-a, namely to point out that many multilingual systems cannot handle texts that are in more than one language.
> - We manually checked that except for “2002-10”,  all the shortest entries are correct abstracts , such as "Éditorial en réponse à un article d'Andrée Quiviger.". We added this in the paper.
> - For papyrus-a, we agree that its usage could be confusing but we believe it is a valuable task for two reasons: 1- it represents a real use case of KPG, since many cultures use code switching heavily while writing and therefore the assumption that only one language is present cannot be made, and 2- it helps point out weaknesses of the current multilingual systems, where language often has to be an input with the sentence, which effectively rule out efficiency for any data that is in more than one language.
>
> As for the distribution, we attempted to plot it but unfortunately a lot of the data has minimum tags (abstracts, keywords, title) and we were therefore unable to extract it in sufficient quantities to get a good idea of the distribution.

---

### Official Review · Reviewer_Tm7j · 2022-07-27
**Review of "A new dataset for multilingual keyphrase generation"**

**Rating:** 6
**Confidence:** 3
**Correctness:** The work seems to be sound and correct.

**Strengths:**

The authors do a great job motivating the need for the multilingual version of keyphrase generation.

Extracting keywords in different languages at the same time (Papyrus-a) is an interesting setup as some important quotes or terms can be written in other languages to preserve their original meaning.


**Weaknesses:**

According to the authors' reference on the percentage of language used on the internet, English and French comprise 29.2% (25.9% + 3.3%) of total languages used. To better reflect the authors' motivation of "multilinguality," other dominant languages such as Chinese and Spanish should be taken into account more in the dataset.

It would be good to mention existing works on multilingual keyphrase extraction as they are also related to the paper's main argument (why not paraphrase those keywords instead of constructing from scratch).


**Additional Feedback:**

N/A

**Clarity:**

The paper is easy-to-follow. Some concerns are noted above.



**Documentation:**

Sufficient documentation has been provided.

**Ethics:**

No.

**Relation To Prior Work:**

Addressed in the section above.

**Summary And Contributions:**

The paper proposes a multilingual keyphrase generation dataset with four versions (Papyrus-e, Papyrus-f, Papyrus-m, Papyrus-a), reflecting multiple scenarios in multilingual keyphrase generation. The original corpus is extracted from a multilingual corpus (Papyrus) that contains various types of documents in different languages. To show the characteristics of the dataset, the authors utilize existing keyphrase extraction methods and BART and discuss some interesting patterns in keyphrase generation in multilingual settings.

---

> ### Author Response · Authors · 2022-08-10
> **Response to "Review of "A new dataset for multilingual keyphrase generation" " (1)**
>
> Thank you for your comments. We agree that the papyrus-m and papyrus-a tasks do not correctly reflect the language diversity of the internet, however we believe that it is a good first step
> and as we mention in the future work we hope to be able to expend the work to include more data in more languages. We also expended the related work to include more existing works on multilingual KPE.
>
> As for paraphrasing, we considered it and tested it briefly but we found the process unreliable. Taking out-of-the-box paraphrasing systems often gives poor results because there is rarely a good way to paraphrase short n-grams. Training a system on top to take extracted keyphrases and transform them is difficult because it implies being able to create links between absent keyphrases and words in the abstract, which is not always possible and difficult to do automatically.

---

### Official Review · Reviewer_PUsF · 2022-07-27
**A multilingual keyphrase dataset with limited analysis**

**Rating:** 4
**Confidence:** 4
**Correctness:** Yes
**Clarity:** Yes

**Strengths:**

Keyphrase generation is valuable task, and having a suitable data set will be helpful to the community.
The method of task data construction sounds reasonable, and the use of data sources is also good

**Weaknesses:**

Lack of detailed explanation of data collection (i.e. the ``heuristic'' in Line 97)
Lack of a more complete introduction to manual evaluation, such as  the quality of evaluators and evaluation criteria
Lack of analysis on the shortcomings of current methods in multilingual scenarios
Although it is a multilingual data set, most of the data are only English and French, and the amount of data in other languages is too small.


**Additional Feedback:**

No

**Documentation:**

Maybe

**Relation To Prior Work:**

Yes

**Summary And Contributions:**

This paper introduces a new multilingual keyphrase dataset and present a BART-based baseline with discussions.

---

> ### Author Response · Authors · 2022-08-10
> **Response to "A multilingual keyphrase dataset with limited analysis"**
>
> Thank you for your comments. We have modified the paper to make it clearer that the heuristic of line 112 is described after. We have also expended on the evaluation protocol, clarifying that it is more of a sanity check by the first author than a formal evaluation. However we'd like to point out that while we have no formal evaluation of the quality of the data and while the amount of data other than French or English is small, results on external multilingual test sets shows that 1- our corpus helps with multilingual KPG, and 2- having a multilingual dataset, even with this small amount of multilingual data, is beneficial over having monolingual corpora on multilingual KPG (Table 5+7). Furthermore, as another reviewer pointed out, the data is uploaded by researchers and checked by the Papyrus team, so there is virtually no chance of erroneous data (the only exception was the "2002-10" which we eliminated, and we added in the paper a precision that we checked that no other mistakes like that happened).

---

### Official Review · Reviewer_HBXh · 2022-07-27
**This paper introducing four new well-formed multilingual datasets on keyphrase generation is very enjoyable.**

**Rating:** 9
**Confidence:** 4
**Clarity:** This paper is well written.

**Strengths:**

•	Significance of contribution: Most of the previous work in the field of KPG is in English. In fact, the only other paper I know of in this specific field that isn’t English was published in 2022. Other work focusing on other languages in the field of keyphrase extraction has been done, but even then, it generally only focuses on a single language. People are just beginning to explore the concept of multilingual KPG, and this is one of the first multilingual KPG datasets.

•	Relevance to the broader research community: KPG can be relevant to many different fields. Augmenting the metadata for a dataset of texts to include keyphrases for each text can help make sorting through that text easier. For example, a dataset that includes the summaries or sneak peaks of books could be augmented to include keyphrases that might help readers choose which book to read.

•	Accessibility and accountability: The authors have clearly laid out how they will keep this dataset up and running and accessible going forward.

•	Ethical and social implications: The ethical implications of this dataset are neutral, but in general I believe that having more data in languages other than English has a positive societal impact because as stated in the paper only around 25% of internet use is in English so that percentage should be reflected in the amount of available data.


**Weaknesses:**

•	Significance of contribution: While two new datasets on this topic have been released this year by Gao et al. in their 2022 paper Retrieval-Augmented Multilingual Keyphrase Generation with Retriever-Generator Iterative Training, their datasets were only released a short time before this dataset would’ve been submitted for review and are significantly different enough that I don’t believe they affect the significance of this paper’s contribution.

**Additional Feedback:**

Have you considered trying other multilingual encoders such as XLM-RoBERTa? Obviously mBART worked well for your purposes, but I’m just curious to know if you tried other methods as well before settling on BART based encoders.

**Correctness:**

To the best of my knowledge the dataset is constructed in a sound way and the claims made in the submission are correct.

**Documentation:**

This dataset is well documented with all necessary information provided in the supplementary materials.

**Ethics:**

No.

**Relation To Prior Work:**

Yes, the authors clearly state how this work relates to and differs from prior work in section 2.

**Summary And Contributions:**

This paper introduces four versions of the Papyrus dataset which focus on keyphrase generation (KPG). The four datasets are Papyrus-e, which includes English examples, Papyrus-f, which includes French examples, Papyrus-m, which includes examples in any language, and Papyrus-a, which includes examples in several different languages at once. The authors also train several different versions of BART on their new datasets and compare results with state-of-the-art models. In general, their models outperform the state-of-the-art models such as YAKE on multilingual KPG.

---

> ### Author Response · Authors · 2022-08-10
> **Response to This paper introducing four new well-formed multilingual datasets on keyphrase generation is very enjoyable.**
>
> Thank you very much for your kind and thoughtful comments. We added discussion of Gao et al. in the Related work section and discussed how the work differs from ours. We are also currently running other types of multilingual encoders-decoders that we hope to integrate in the final submission. We settled on mBART because we followed Chowdhury et al. for our protocol, but we agree that it would be best to have various systems.

---

### Official Review · Reviewer_CCHh · 2022-07-28
**The dataset is important, but it needs refinements.**

**Rating:** 5
**Confidence:** 3
**Correctness:** Yes.
**Clarity:** Yes, the paper is well written and ea…

**Strengths:**

The task and dataset description of this paper is clear. The multilingual corpus comes from institutional reports or documents written by faculty members, therefore, the dataset quality can be relatively high.

**Weaknesses:**

1. As a new dataset, data quality control or data quality evaluation is needed.
2. As a dataset for multilingual keyphrase generation, languages presented in the dataset are rather imbalanced. Except for English and French data, the amount of abstracts in other languages is too small.

**Additional Feedback:**

None.

**Documentation:**

Documentation and a hosting, licensing and maintenance plan are expected.

**Ethics:**

No ethical discussion is provided by the authors. It would be desirable for authors to address it.

**Relation To Prior Work:**

The discussions in the related work part are not comprehensive enough. The authors clarify the difference with Kp20K in terms of statistical information and baseline model performance.

**Summary And Contributions:**

The paper proposes a new dataset for multilingual keyphrase generation, Papyrus, which includes 16,247 pairs of abstracts and key phrases. The multilingual corpus is collected from Papyrus, a repository of documents from the Université de Montréal.

---

> ### Author Response · Authors · 2022-08-10
> **Response to "The dataset is important, but it needs refinements"**
>
> Thank you very much for your time and thoughtful comments.
>
> On data quality control: We sanity checked our process by randomly sampling and checking 100 examples (this is described L122-L125). As another reviewer pointed out, the data is uploaded by researchers and checked by the Papyrus team, so there is virtually no chance of erroneous data (the only exception we found was the "2002-10" which we eliminated, and we added in the paper a precision that we checked that no other similar mistakes happened). Do you have a more elaborated evaluation procedure to suggest?
> On imbalance of data: We agree that this is a weakness of the papyrus-m and papyrus-a tasks. However we'd like to point out that even this small amount of data is useful for multilingual KPG, as we show in Table 5 on test sets of various languages, and we show in Table 7 that the system learn to generate keyphrases independently of language. We would eventually like the dataset to keep expending to add more languages from other university repositories but we believe this is a solid first step. We actually considered keeping only French and English initially, but ended up considering that even if we were to expend the dataset, some languages are still going to be present in small quantities (ex Inuktitut) and we think that removing the long tail would just participate in the lack of data for rare languages in KPG and ML in general.
> On Prior work: We extended the related work section to include more sources, especially relating to more KPG and KPE works in other languages. Thank you for pointing out the lack of references to us.
> On Ethics: We discuss the ethics of the data collection in the joined datasheet for dataset, but thank you for pointing that the discussion was lacking in the paper. We have added a more thorough discussion in the “Implications” section.

---

### Official Review · Reviewer_SPa3 · 2022-07-28
**Good paper, useful dataset**

**Rating:** 6
**Confidence:** 4
**Correctness:** The dataset details and baseline metr…
**Clarity:** The paper is well structured and easy…

**Strengths:**

(1) The methods used for data collection are simple and straightforward including tools used for language identification. It is also very easy to extend this to document libraries outside of Papyrus with little manual intervention.
(2) Papyrus-f appears to be very useful for evaluating French KPG models.


**Weaknesses:**

The theme of the paper seems to be constantly switching between French and other languages. Other languages are mentioned quite casually with no details on how or why the language distribution is that way. Another detail missing is if the language distribution was used while training the multilingual models or while evaluating them (looks like it wasn’t). It is also a little hard for me to swallow that French itself is a minority language given the vast amount of literature in MT and unsupervised pretraining in the language. I would urge the authors to reconsider the wording in Section 6.


**Additional Feedback:**

L160: Citation style is incorrect, should be [38,28,26]
L264: imbalance* of data
Table 6: language codes need to be expanded somewhere in the paper


**Documentation:**

Authors provide a link to a github repository in the paper containing data, scripts and instructions to get started.

**Ethics:**

None. The data is collected from Papyrus: a publicly available index of documents.

**Relation To Prior Work:**

Relation to prior work is discussed adequately. Maybe include using pretrained T5 models for comparison since they also tend to do very well on summarization/abstraction tasks?

**Summary And Contributions:**

This paper presents the first dataset focused on keyphrase generation for the French language. Previous works usually relied on unsupervised methods to perform extractive keyphrase generations. The authors also release several baseline models on related tasks (Papyrus f/e/m/a and a few previous datasets) that are useful for comparison.

---

> ### Author Response · Authors · 2022-08-10
> **Response to "Good paper, Useful Dataset"**
>
> Thank you for your thoughtful comments. We corrected the uploaded version accordingly (including reformulating section 6). You are right in saying that French is not a minority language. Our intent was to warn that the classic metrics (F1, R, P@K) can erase the performance on languages that are rare in the corpus (languages other than French and English). We clarified that a better way to test the system was to compute metrics per language as well as test on downstream tasks.
>
> We made clear that the language distribution was not used in training nor in evaluating. We are not familiar with any techniques that would help alleviate the skewed distribution in this specific case. For evaluation, instead of trying to adapt the metrics to the long-tail situation and possibly create results that are difficult to compare to other systems, we output the results on the individual languages present in the test set.
>
> We expended the language codes of Table 6 in the caption.
>
>  On a note for the bibliography style of line of L160, the neurips file said we could use any citation style as long as it was consistent. We elected to follow the acl_natbib style which format with ; in \citep context. We hope that this is ok, please let us know if we missed any instructions that said otherwise.
>
> Finally, we are currently, as you suggested, working on an implementation of T5 for which we hope to have results in time for the final submission. We chose BART because we followed the paper of Chowdhury et al. for implementing and comparing results, but it is true that T5 has shown very good results and should be tested as well. Thank you.

---

### Meta-Review · Area_Chair_dSf4 · 2022-09-09

**Recommendation:** Accept
**Confidence:** 4

**Metareview:**

This paper presents the dataset Papyrus, consisting of abstracts and keyphrases extracted from institutional reports from the Université de Montréal. The dataset is mostly in two languages and was generated using straightforward, yet easily extensible methods for extraction. This is one of the first datasets that expands KPG beyond English. Although limited in coverage, it sets the tone for generalization of techniques beyond English.


**pros**
* Expands the field of KPG to other languages beyond English.
* The methods for extraction are straightforward.
* Given its origin, the quality of the data is expected to be high.
* A portion of the data has been validated through human evaluation.

**cons**
* The coverage of the dataset is mostly limited to English and French. Other languages aren't well represented.

---

### Decision · Program_Chairs · 2022-09-16

Accept